# Mutational spectra are associated with bacterial niche

Christopher Ruis [1,2,3], Aaron Weimann[1,2,3], Gerry Tonkin-Hill[4], Arun Prasad Pandurangan[5], Marta Matuszewska[2,6], Gemma G. R. Murray [7], Roger C. Lévesque [8], Tom L. Blundell [5], R. Andres Floto [1,3,9,10] ✉ & Julian Parkhill [2,10] ✉

As observed in cancers, individual mutagens and defects in DNA repair create distinctive mutational signatures that combine to form context-specific spectra within cells. We reasoned that similar processes must occur in bacterial lineages, potentially allowing decomposition analysis to detect both disruption of DNA repair processes and exposure to niche-specific mutagens. Here we reconstruct mutational spectra for 84 clades from 31 diverse bacterial species and find distinct mutational patterns. We extract signatures driven by specific DNA repair defects using hypermutator lineages, and further deconvolute the spectra into multiple signatures operating within different clades. We show that these signatures are explained by both bacterial phylogeny and replication niche. By comparing mutational spectra of clades from different environmental and biological locations, we identify niche-associated mutational signatures, and then employ these signatures to infer the predominant replication niches for several clades where this was previously obscure. Our results show that mutational spectra may be associated with sites of bacterial replication when mutagen exposures differ, and can be used in these cases to infer transmission routes for established and emergent human bacterial pathogens.

Studies on human cells and tissues have demonstrated that mutagens, endogenous mutagenesis, and defects in DNA repair each induce highly specific context-dependent patterns of base substitutions termed mutational signatures, which combine to form a mutational spectrum[1–9]. Reconstructing the set of mutations and signatures within cancers has enabled inference of the drivers of tumourigenesis[1,2,7]. We therefore reasoned that reconstructing mutational spectra in bacteria, decomposing them into different signatures, and correlating these with known DNA repair defects and environmental exposures, should

allow the association of specific signatures with bacterial replication niches. We define niches as replication sites within which the bacterium spends sufficient time to be exposed to mutagens; depending on the bacterium, this may be a mixture of colonization and infection sites, colonization sites only or infection sites only. These signatures could potentially be used both to infer environmental location or site of colonization and/or infection and to identify defects in DNA repair when niche is known. In this work, we tested this by undertaking a large-scale comparison of mutational spectra and their underlying

[1]Molecular Immunity Unit, University of Cambridge Department of Medicine, MRC-Laboratory of Molecular Biology, Cambridge, UK. [2]Department of Veterinary Medicine, University of Cambridge, Cambridge, UK. [3]Cambridge Centre for AI in Medicine, University of Cambridge, Cambridge, UK. [4]Department of Biostatistics, University of Oslo, Blindern, Norway. [5]Department of Biochemistry, Sanger Building, University of Cambridge, Cambridge, UK. [6]Department of Medicine, University of Cambridge, Cambridge, UK. [7]Parasites and Microbes Programme, Wellcome Sanger Institute; Wellcome Genome Campus, Cambridge, UK. [8]Institut de biologie intégrative et des systèmes (IBIS), Université Laval, Québec City, Québec, Canada. [9]Cambridge Centre for Lung Infection, Papworth Hospital, Cambridge, UK. [10]These authors contributed equally: R. Andres Floto, Julian Parkhill. ✉e-mail: arf27@cam.ac.uk; jp369@cam.ac.uk

signatures across bacteria, correlating the results with specific DNA repair defects and replication niche. We identify mutational signatures associated with defects in DNA repair and with replication niche, and apply these niche signatures to infer transmission routes for several bacterial clades where this was previously unclear.

## Results and discussion
### Bacteria exhibit diverse mutational spectra
Using a specifically-developed open-source bioinformatic tool, *MutTui* (https://github.com/chrisruis/MutTui), we analysed whole genome sequence alignments and phylogenetic trees to reconstruct single base substitution (SBS) mutational spectra of 84 phylogenetic clades from 31 diverse bacterial species representing a broad range of phylogenetic diversity and replication sites (Figs. S1 and S2; Supplementary Data 1 and 2; Methods; dataset sources and replication sites are described in Supplementary Note 1). SBS spectra were rescaled by genomic nucleotide composition to enable direct comparison between bacteria. We find that SBS spectra are highly diverse, both in the nucleotide mutations themselves and their surrounding context (Figs. 1 and S2).

Using this approach, several generalisable properties of bacterial SBS spectra could be identified. As expected from previous analysis[10], we found that transition mutations are more common than transversion mutations in all cases (ranging from 52 to 55% in *Klebsiella pneumoniae* to >90% in *Campylobacter jejuni*; Fig. S2) and cytosine to thymine (C > T) was typically the most common mutation type identified (in 69 of 84 SBS spectra examined), potentially due to cytosine deamination[11]. T > C was the most common mutation type in the remaining 15 spectra. Genomic G + C content exhibited a negative correlation with proportion of C > A/T mutations but a positive correlation with proportion of C > G mutations (Fig. S3). Finally, transition mutations exhibit enriched context specificity compared to transversion mutations while several contextual mutations are significantly elevated across datasets (Fig. S4).

We observe a strong correlation between phylogenetic relatedness and spectrum similarity (Tukey HSD corrected ANOVA $P < 0.001$; Fig. S5), with spectra being typically conserved across highly related clades in which there has likely been no change of niche or DNA repair capacity (Figs. 1 and S6). Since these spectra represent composites of mutagenesis and DNA repair, we reasoned that they could be decomposed into combinations of specific mutational signatures, each driven by distinct defects in DNA repair, by endogenous processes, or by specific mutagens[12], as has previously been achieved for cancer-associated spectra[1,3,7].

### DNA repair genes are associated with diverse mutational signatures
We first used natural variation to extract contextual mutational signatures associated with a broad range of distinct DNA repair pathways by calculating SBS spectra of 50 naturally occurring hypermutator lineages across four bacterial species (Fig. 2A, Fig. S7). We identified mutations and frameshifts in DNA repair genes that occurred on, or immediately ancestral to, each hypermutator lineage (Methods) to infer the genes most likely responsible for hypermutation, enabling us to attribute mutational signatures to defects in 11 DNA repair genes that function in mismatch repair (MMR), base excision repair (BER), or homologous recombination (HR) (Fig. 2B–D). Although mutational types associated with knockouts of several DNA repair genes have previously been characterised in vitro in a limited number of bacterial species[10,13–18], contextual signatures (which increase the power to discriminate between distinct mutational drivers) have only previously been calculated for MMR defects in *Escherichia coli* and *Pseudomonas aeruginosa*[10,13].

Naturally occurring deleterious mutations of bacterial MMR genes result in high levels of context-specific C > T and T > C mutations (Figs. 2B and S8)[1,8,19], similar to previously calculated in vitro signatures[10,13,15,16] and likely represent the error profile of DNA Polymerase III that is usually repaired by functional MMR. T > C mutations

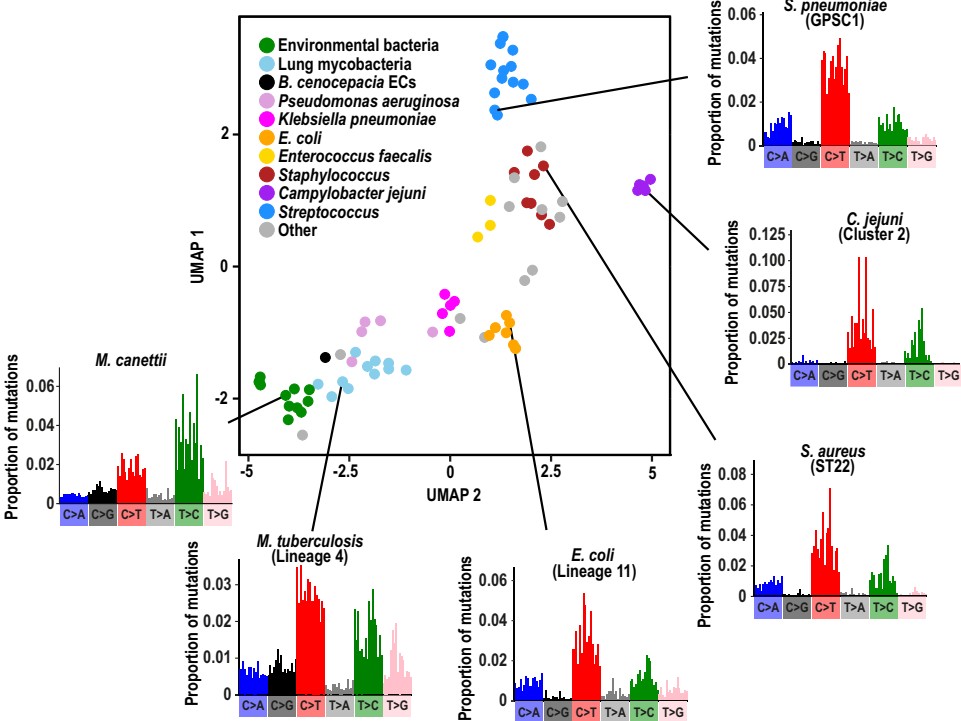

**Fig. 1 | Clustering of bacterial SBS spectra.** UMAP clustering based on contextual mutation proportions within the 84 SBS spectra across 31 bacterial species. Selected groups are coloured. The environmental bacteria label includes *Burkholderia pseudomallei* and known environmental *Mycobacteria*. Example SBS spectra are shown for selected groups. Source data are provided as a Source Data file.

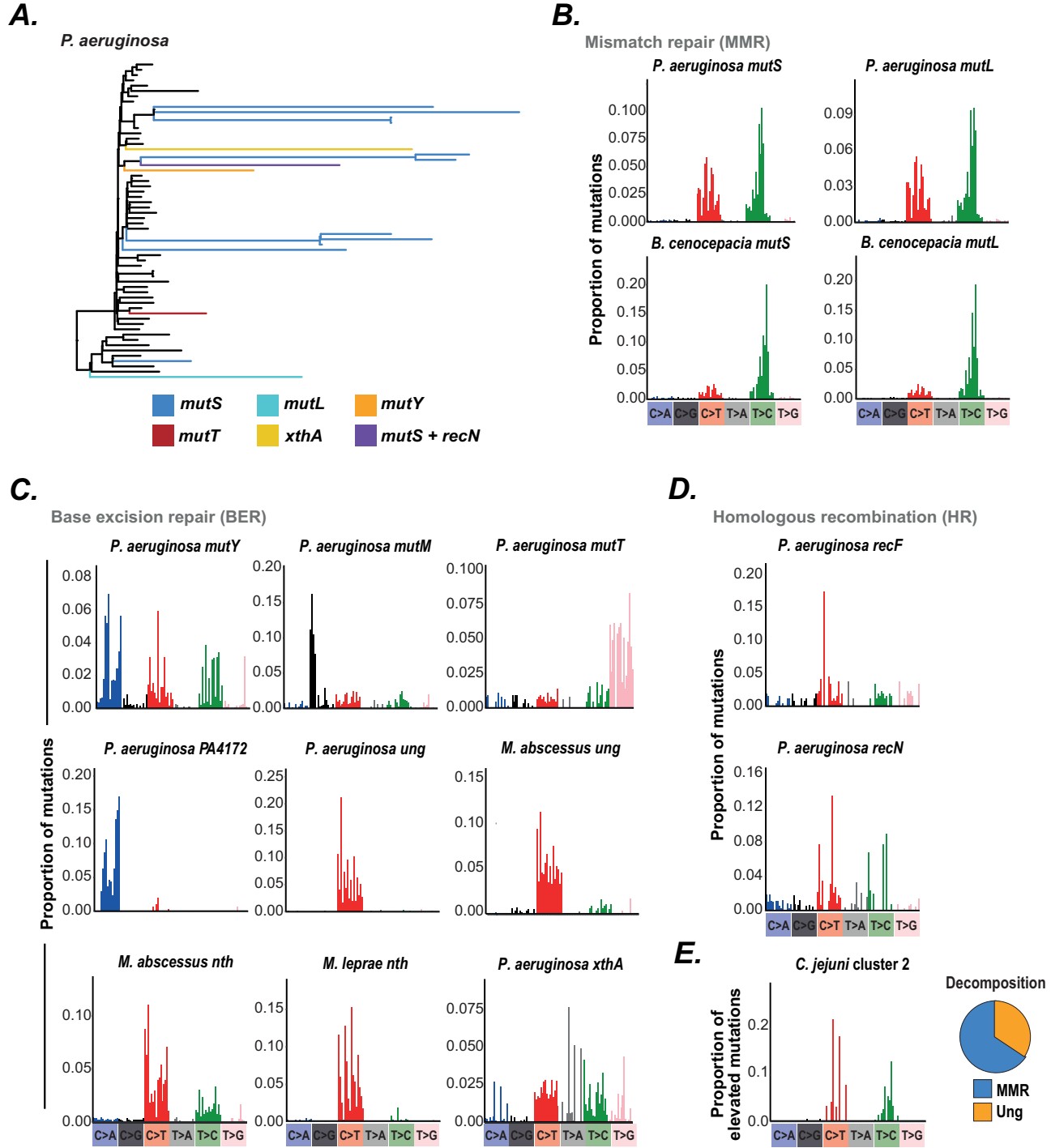

**Fig. 2 | Mutational signatures associated with DNA repair genes. A** Example *P. aeruginosa* phylogenetic tree (ST274) showing hypermutator branches and the inferred responsible genes. Hypermutator branches were identified based on branch length and the ratio of transition and transversion mutations. Responsible genes were identified as DNA repair genes exhibiting a mutation on the long phylogenetic branch or ancestral branch. Black branches are background non-hypermutator branches that did not contribute to hypermutator spectra. **B** Mutational signatures associated with MMR genes. **C** Mutational signatures

associated with BER genes. **D** Mutational signatures associated with genes involved in homologous recombination. **E** Top panel shows the mutations elevated in *C. jejuni* cluster 2 compared with *E. coli* lineage 34, calculated by subtracting each respective mutation proportion in the SBS spectra. The pie chart shows the proportion of mutations elevated in *C. jejuni* cluster 2 that are assigned to each bacterial DNA repair gene signature in a decomposition analysis. Source data are provided as a Source Data file.

are particularly enriched in GpTpN contexts in both *P. aeruginosa* and *Burkholderia cenocepacia* (Fig. 2B), similar to previous in vitro results[10,13]. While context specificity is highly similar between species, the relative rates of C > T and T > C differ between *P. aeruginosa* and *B. cenocepacia* (Figs. 2B and S9), likely reflecting distinct polymerase

error profiles (a possibility supported by structural modelling analysis; Fig. S10).

Mutations in distinct base excision repair (BER) components result in characteristic gene-specific patterns (Fig. 2C), as expected from the diverse repair functions of proteins within this pathway[11].

We identified *P. aeruginosa* hypermutators for each component of the GO repair pathway (*mutT*, *mutY* and *mutM*) that prevents 8-oxoguanine (8-oxo-G)-induced mutations[20]. Mutation of *mutT*, whose product degrades 8-oxo-G monomers to prevent their incorporation into DNA[20], results in non-specific T > G mutations (Fig. 2C), suggesting incorporation of 8-oxo-G opposite adenine is context-independent. Conversely, mutation of *mutY* which excises adenine opposite 8-oxo-G[20], results in C > A mutations predominantly in CpCpN and TpCpN contexts (Fig. 2C), indicating context-specific mutation of incorporated guanine to 8-oxo-G. This likely represents the pattern of reactive oxygen species (ROS) damage, of which 8-oxo-G is a major mutagenic lesion[9]. The C > A contexts differ between the *P. aeruginosa mutY* signature and human cell signatures of ROS exposure[5] and knockout of either the *mutY* homologue or OGG1[7,9] (Fig. S11), suggesting differential repair of these lesions by other proteins. Mutation of *mutM* results in C > G mutations in ApCpN contexts (Fig. 2C). While the mechanism of C > G mutations is unclear, the lack of C > A mutations in *mutM* knockouts is potentially due to functional MutY being sufficient to repair mutagenic 8-oxo-G lesions in *P. aeruginosa*[21]. We additionally identify PA4172 in *P. aeruginosa* whose knockout exhibits C > A mutations in CpCpN and TpCpN contexts similar to *mutY* (Pearson's *r P* < 0.001; Fig. 2C; Fig. S11), suggesting that its product may similarly repair mutagenic 8-oxo-G lesions.

Disruption of *ung*, whose product removes uracil from DNA[11], results in similar patterns of context-specific C > T mutations in *P. aeruginosa* and *Mycobacterium abscessus* (Pearson's *r P* < 0.001; Fig. 2C; Fig. S12). This bacterial signature exhibits subtle contextual differences compared with *ung* knockout in human cells[9], particularly through enriched mutations in NpCpG contexts (Fig. S12), suggesting differential patterns of uracil incorporation in humans and bacteria.

Mutation of *nth*, whose product Endonuclease III removes damaged pyrimidines, results in C > T mutations in multiple *Mycobacteria* species and human cells[8] but with different context specificity (Pearson's *r P* > 0.05; Fig. 2C; Fig. S13). Disruption of the apurinic-apyrimidinic (AP) endonuclease *xthA* results in mutations in multiple specific contexts (Fig. 2C), particularly transversions in [C,G,T]p[C,T] pG contexts, indicating repair of a broad range of specific lesions. Finally, hypermutators resulting from mutation of the homologous recombination pathway components *recF* and *recN* exhibit context-specific transition mutations (Fig. 2D). Recombination is known to drive GC-biased gene conversion[22] and this may contribute to this signature.

We next examined whether different bacterial species that occupy a similar niche (and are therefore exposed to similar sets of niche-specific mutagens) might exhibit spectrum differences consistent with differences in DNA repair. *Campylobacter jejuni* has previously been shown to be deficient in several types of DNA repair[23] so we compared mutations accumulated by *C. jejuni* with those accumulated in the gastrointestinal *E. coli* lineage 34[24]. Decomposition analysis showed that almost all mutations elevated in *C. jejuni* could be explained by a failure to repair deaminated cytosines and a lack of MMR (Fig. 2E; Fig. S14); pathways which are known to be absent in *C. jejuni*[23]. These results indicate that differences in DNA repair may be inferred by comparing bacteria from a similar niche.

## Bacteria exhibit phylogeny-associated and non-phylogeny-associated mutational signatures

We then proceeded to extract de novo further bacterial signatures through a decomposition analysis employing non-negative matrix factorisation (NMF)[25,26] on SBS spectra datasets from a range of species and genera (Supplementary Data 3). We extracted 33 SBS signatures and collapsed these into a final set of 24 (named with the prefix Bacteria_SBS) by combining highly similar signatures (with cosine similarity of 0.95 or greater) (Fig. S15; Supplementary Data 4). The

extracted signatures exhibit divergent base mutations and contexts (Fig. 3A). As these signatures predominantly consist of multiple mutation types, they are likely to be composites formed from the action of multiple underlying mutagenic and/or repair processes. Most signatures are genus-specific (Fig. 3B), supporting differential activity of mutagens and repair between clades and a strong influence of bacterial phylogeny on mutational patterns. An exception to this was signature Bacteria_SBS15 that was extracted from the *Staphylococcus* genus, *Enterococcus faecalis*, *Streptococcus pneumoniae* and *Streptococcus agalactiae* datasets (Fig. 3B), indicating broad distribution across Bacillota. As these bacteria inhabit different niches, this signature likely represents phylum-specific endogenous mutations and/or DNA repair profiles.

Most SBS spectra were decomposed into multiple signatures. Comparison of signature clustering with bacterial phylogeny showed that the similarity between signatures does not follow phylogenetic relationships, with interspersion of extracted signatures across the deeper parts of the bacterial tree and multiple examples of similar signatures extracted from diverse bacterial genera (Fig. 3B). For example, Bacteria_SBS1 was extracted from *Mycobacteria* and is similar to Bacteria_SBS13 extracted from *Burkholderia*, while Bacteria_SBS6 within *E. coli* is similar to the streptococcal signature Bacteria_SBS18. These results support a significant role for additional non-phylogeny factors in shaping mutational spectra. We hypothesised that niche-specific mutagens are one such factor that may act on multiple bacterial clades inhabiting similar niches. We therefore tested the contribution of niche to mutational spectra by comparing linear models incorporating niche and genus with models including genus alone. We examined the fit of these models to mutation type proportions across the full dataset (82 clades, as we excluded two clades where niche is controversial) and found that incorporating niche resulted in a significantly better fit for four of the six mutation types: C > A, C > T, T > A and T > C (Fig. 3C, *p* < 0.005). Furthermore, including niche significantly improved fit to 92 of the 96 contextual mutations (Supplementary Data 5, *p* < 0.05). While species-level signatures may still contribute (and our analysis includes multiple clades that share the same niche jump, and may therefore not represent individual events), these results strongly suggest that niche plays a significant role in shaping mutational spectra.

## Replication niches can drive mutational signatures

We therefore aimed to identify mutational signatures associated with specific niches, initially focussing on *Mycobacteria* and *Burkholderia* as these genera contain multiple independent comparisons between clades that are transmitted from person-to-person and clades that are acquired from environmental sources[27,28]. We found that known lung and environmental clades cluster separately based on SBS spectrum composition (Fig. 4A). Spectrum subtractions consistently revealed elevated C > A[29] and C > T mutations in lung bacteria and higher levels of T > C in environmental bacteria (Fig. 4A, Fig. S16). We correspondingly observed that including a binary lung or environmental niche variable significantly improves the fit of linear models to C > A, C > T and T > C proportions compared to a model including genus only (Fig. S17). Lung and environmental bacteria additionally exhibit different contextual patterns within C > A and T > C mutations (Fig. S18). Decomposition of niche-specific mutations from subtracted spectra using known human mutagen signatures suggests that higher C > A mutations in lung bacteria may be driven by exposure to mutagens including reactive oxygen species (ROS), while higher T > C mutations within the environment is potentially caused by exposure to alkylating agents and nitro-polycyclic aromatic hydrocarbons (Fig. 4B), both established environmental mutagens[5]. As this decomposition analysis can only identify mutagens with known signatures, additional mutagens with currently uncharacterised signatures may contribute to the observed patterns. It is also possible that the long-term evolutionary

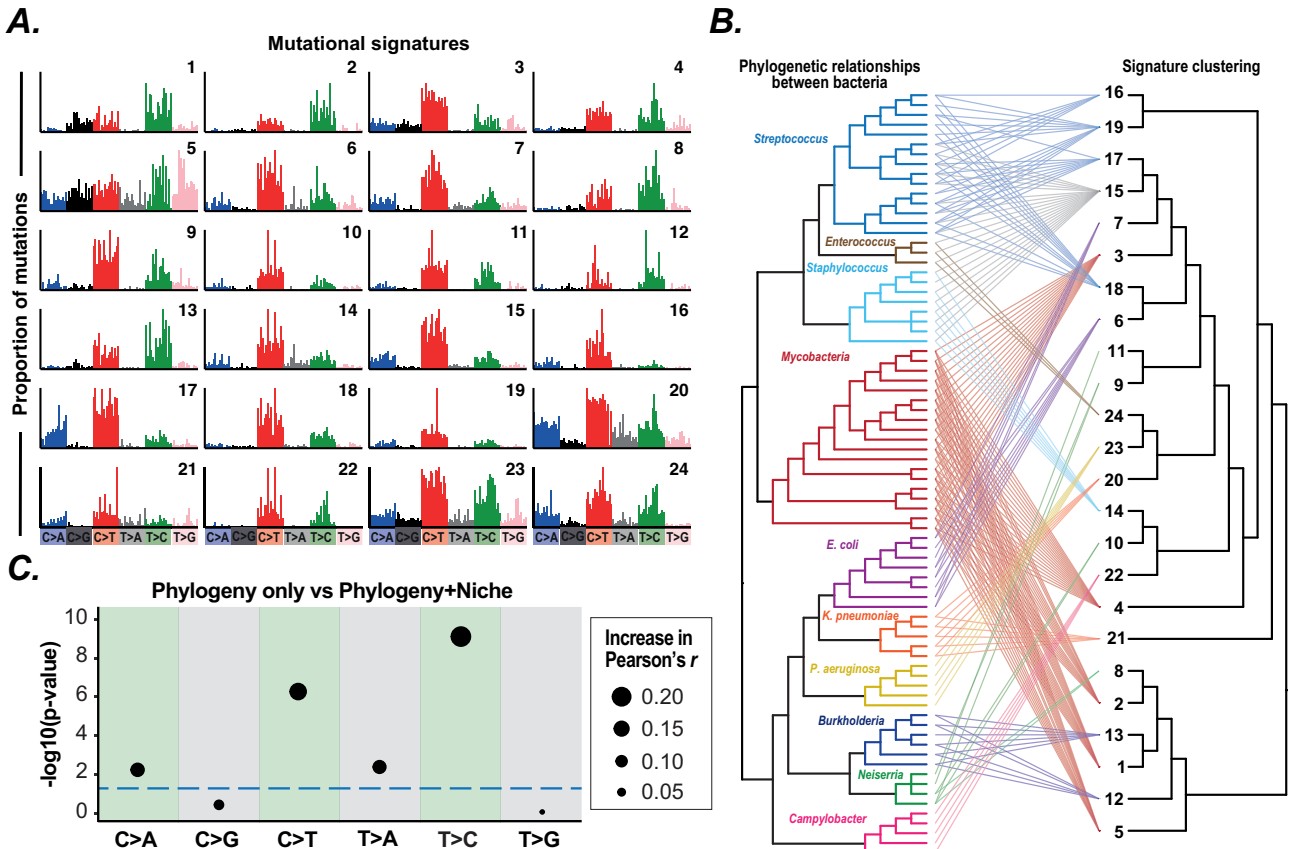

**Fig. 3 | Pathogen mutational spectra are influenced by niche. A** SBS spectra for the 24 mutational signatures extracted de novo, showing the proportion of each contextual mutation. **B** Tanglegram showing the relationship between bacterial phylogeny (left hand tree) and mutational signature clustering (right hand tree). Connecting lines show that the composite signature was identified in the corresponding SBS spectrum in at least one signature extraction. Lines are coloured to match the colour of the source genus in the phylogeny, with the exception of signature 15 which was identified in multiple genera and is coloured grey. **C** We

compared the ability of a linear model incorporating genus only and a linear model including both genus and niche to explain mutation type proportions across SBS spectra. LOD scores assessing significance of the improvement in model fit for the model including both genus and niche over the genus only model are shown. Model fits were assessed for each mutation type separately. Points are sized by the improvement in Pearson's r correlation when including niche over the genus only model. Blue dashed line indicates $p = 0.05$; points above this line show a significant improvement in model fit. Source data are provided as a Source Data file.

selection towards GC richness seen in some bacterial genomes[30] may contribute to the observed environmental signature. Nevertheless our findings provide strong support for the ability to extract information from spectra about bacterial exposure to niche-specific mutagens[31] and supports a central role for ROS in host immunity to these pathogens (as suggested by previous experimental studies[32,33]).

We further examined niche signatures through a targeted NMF decomposition of the *Mycobacteria* and *Burkholderia* spectra and were able to extract a human lung-associated mutational signature consisting of multiple mutation types that we term Bacteria_Lung1 (Fig. 4C, D). This signature is elevated in human lung clades with independent origins across the *Mycobacteria* and *Burkholderia* phylogeny (Fig. 4E).

### Inference of Mycobacteria replication niches from mutational patterns

Due to the separation between known niches, we next used SBS spectra to infer niche for two *Mycobacteria* clades where this was previously poorly understood. The dominant circulating clones (DCCs) of *M. abscessus* have emerged as an important global cause of pulmonary infections in individuals with Cystic Fibrosis (CF) and other lung conditions[27,34]. While whole genome sequencing has shown that many geographically widespread patients are infected with near-identical bacteria[27,34], epidemiological linkage typically cannot be established between such patients. This has led to the niche(s) and transmission pathways of the DCCs being controversial, with some

studies suggesting most cases arise through human-to-human lung transmission[27,35] while other studies suggest independent acquisition from environmental reservoirs[36–38]. The mutational spectrum of the DCCs strongly suggests that they are replicating within, and transmitting from, the lung since they: cluster with known human lung bacteria based on the SBS spectrum (Fig. 4A); exhibit lung-like contextual patterns of C > A and T > C mutations (Fig. S18); exhibit high levels of C > A and low levels of T > C (Fig. 4A); and exhibit signature Bacteria_Lung1 at similar levels to known lung bacteria (Fig. 4D). While it is possible that other body sites could exhibit similar mutational patterns, our observations strongly suggest that most DCC infections are acquired through human-to-human lung transmission.

The *Mycobacterium kansasii* main cluster (MKMC) causes the majority of *M. kansasii* lung infections[39] and was previously thought to be independently acquired from water sources[39]. However, the MKMC spectrum exhibits characteristics of both lung and environmental spectra. Specifically, the MKMC exhibits lung-like C > A patterns but environmental-like T > C patterns (Fig. S18) and is therefore intermediate between known human lung and environmental spectra in the SBS clustering and C > A vs T > C comparison (Fig. 4A). Together, these results suggest that the MKMC is exposed to both lung and environmental mutagens and therefore likely replicates within (and is potentially acquired from) both niches. In further support of this, we find close international transmission linkages that are likely driven by human-to-human transmission (Fig. S19, Supplementary Note 2), and

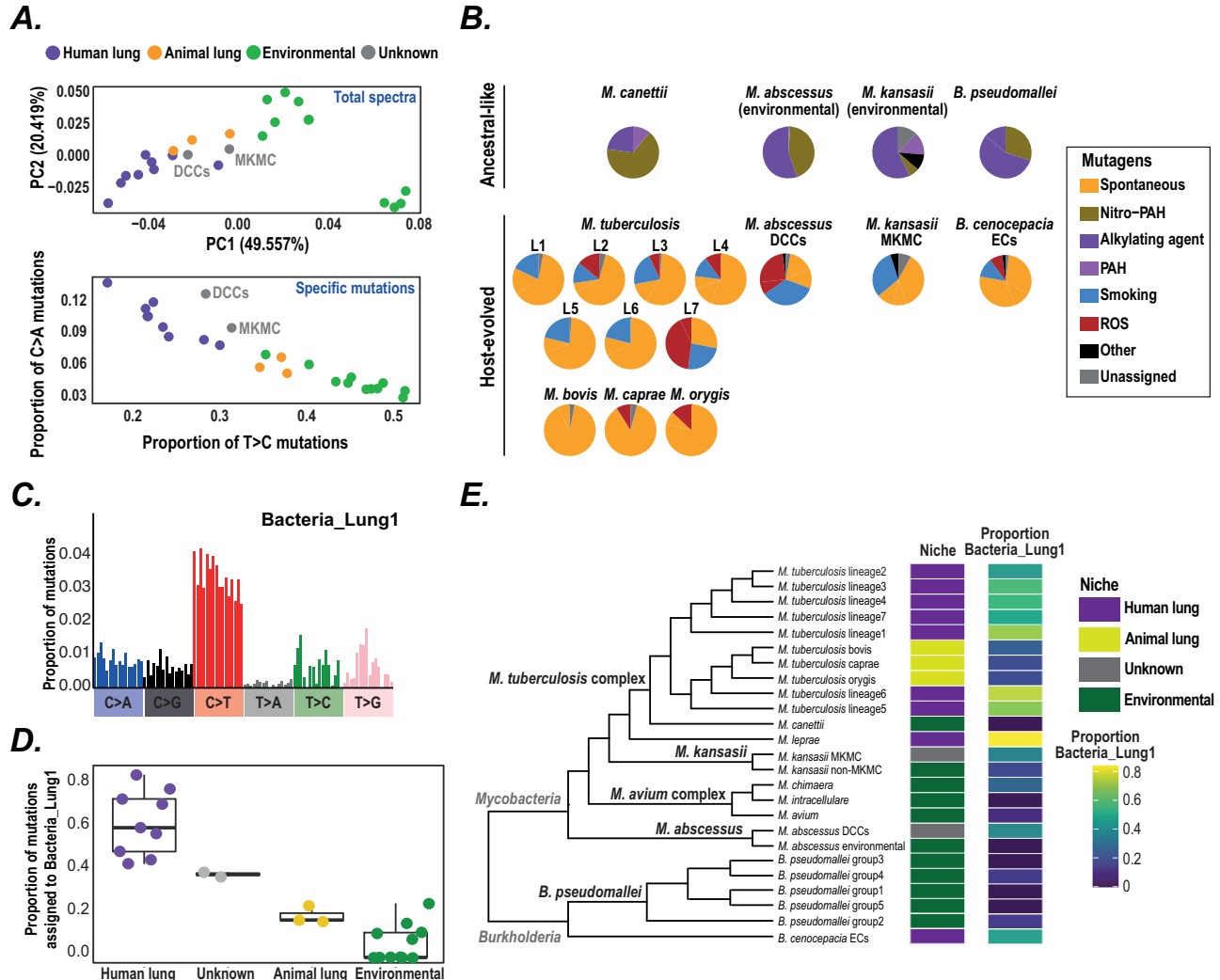

**Fig. 4 | Comparison of mutational spectra between lung and environmental niches. A** Upper panel - principal component analysis on mutation proportions in the SBS spectra across *Mycobacteria* and *Burkholderia*. Axes labels include the inferred proportion of variance each principal component describes. Points are coloured by niche; clades with a previously unknown niche are labelled. Environmental includes *B. pseudomallei* and known environmental clades of *Mycobacteria*. Lower panel - comparison of the proportion of T > C and proportion of C > A mutations in *Mycobacteria* and *Burkholderia* SBS spectra. **B** Decomposition of mutational spectra into their underlying components. Only mutations elevated within the respective clade compared to a closely related clade in a different niche were included. Known environmental clades were decomposed into the set of previously extracted environmental mutagen signatures[5] while known lung clades and clades with unknown niche were decomposed into the set of previously extracted lung signatures from human data. *B. cenocepacia* ECs: *B. cenocepacia* epidemic clones. Nitro-PAH: nitro-polycyclic aromatic hydrocarbons; PAH: polycyclic aromatic hydrocarbons; ROS: reactive oxygen species. **C** Composition of signature Bacteria_Lung1 extracted from NMF decomposition of *Mycobacteria* and *Burkholderia* SBS spectra. **D** The proportion of mutations within each *Mycobacteria* and *Burkholderia* SBS spectrum assigned to signature Bacteria_Lung1. Boxplot centre lines show median value; upper and lower bounds show the 25th and 75th quantile, respectively; upper and lower whiskers show the largest and smallest values within 1.5 times the interquartile range above the 75th percentile and below the 25th percentile, respectively. All clade values are shown as points (number of clades included: Human lung = 9, Unknown = 2, Animal lung = 3, Environmental = 11). **E** Dendrogram shows phylogenetic relationships between *Mycobacteria* and *Burkholderia*. The left hand heatmap shows niche of each clade; lung clades have arisen on multiple independent occasions across the tree. The right hand heatmap shows the proportion of mutations assigned to signature Bacteria_Lung1 in a decomposition analysis of the *Mycobacteria* and *Burkholderia* spectra. More mutations are consistently assigned to Bacteria_Lung1 in lung clades than environmental clades and lung clades exhibit a higher assignment to Bacteria_Lung1 than closely related environmental clades. Source data are provided as a Source Data file.

identify mutations of known lung pathoadaptive genes[40] on internal phylogenetic branches that transmit to multiple patients and exhibit characteristics of lung mutational spectra (Fig. S20, Fig. S21, Supplementary Note 2). These observations support human-to-human transmission being a major contributor to MKMC infections.

### Identification of additional niche-associated mutational signatures

Finally, we extended our approach to identify mutational signatures associated with other bacterial replication niches in humans. We used further linear models to test for signatures associated with the upper respiratory tract (URT) where we have independent pairs of URT and non-URT clades across three genera: *Staphylococcus*, *Streptococcus* and *Neisseria*. This identified reduced T > A mutations associated with URT replication in each genus (Fig. 5A, Fig. S22) (likely due to a higher rate of T > A mutations in non-URT niches). We also tested for differences in spectrum between bacteria with different micro-niche preferences in human skin. We find a high level of CC > TT double mutations characteristic of UV-light damage[5] in the pan-skin bacterium *Cutibacterium acnes* that is not present in *Staphylococcus epidermidis*

**A.**

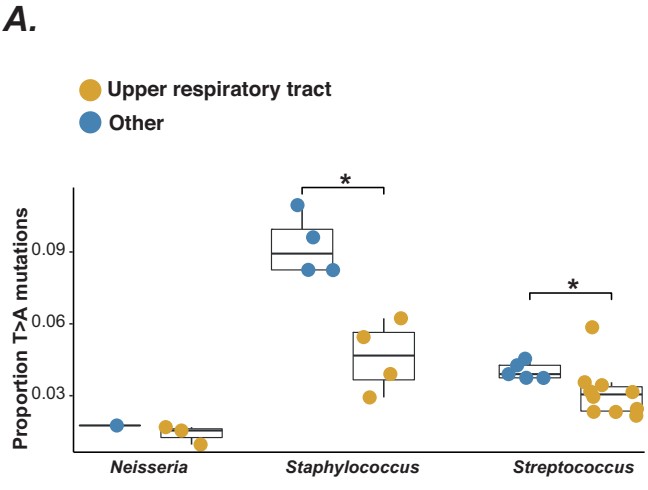

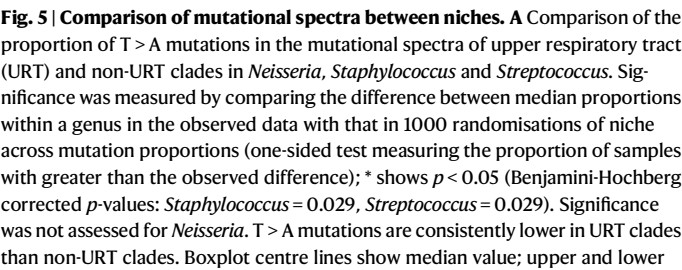

**B.**

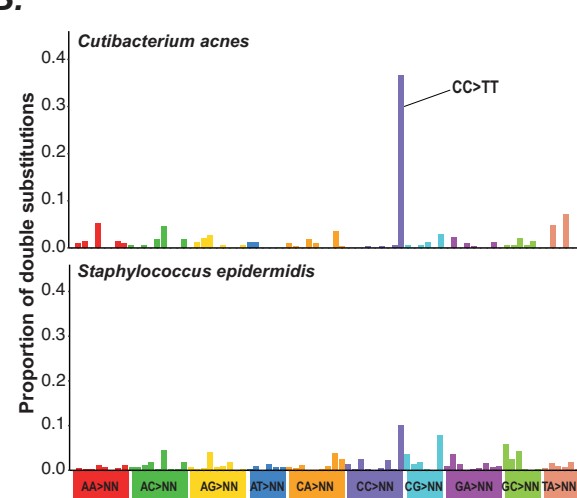

**Fig. 5 | Comparison of mutational spectra between niches. A** Comparison of the proportion of T > A mutations in the mutational spectra of upper respiratory tract (URT) and non-URT clades in *Neisseria*, *Staphylococcus* and *Streptococcus*. Significance was measured by comparing the difference between median proportions within a genus in the observed data with that in 1000 randomisations of niche across mutation proportions (one-sided test measuring the proportion of samples with greater than the observed difference); * shows $p < 0.05$ (Benjamini-Hochberg corrected $p$-values: *Staphylococcus* = 0.029, *Streptococcus* = 0.029). Significance was not assessed for *Neisseria*. T > A mutations are consistently lower in URT clades than non-URT clades. Boxplot centre lines show median value; upper and lower

bounds show the 25th and 75th quantile, respectively; upper and lower whiskers show the largest and smallest values within 1.5 times the interquartile range above the 75th percentile and below the 25th percentile, respectively. All clade values are shown as points (number of clades included: *Neisseria* other = 1, *Neisseria* URT = 3, *Staphylococcus* other = 4, *Staphylococcus* URT = 4, *Streptococcus* other = 5, *Streptococcus* URT = 10). **B** Comparison of double substitution spectra between *C. acnes* and *S. epidermidis* (phylogenetic groups A-C combined). CC > TT is indicated in the *C. acnes* spectrum and is a classic signature of exposure to UV light. Source data are provided as a Source Data file.

which preferentially inhabits moist, and therefore less sun-exposed, skin sites such as the groin and armpits[41] (Fig. 5B). This again suggests that niche-associated differences in mutagen exposure can leave detectable signatures in mutational spectra.

In conclusion, we show that we can reconstruct mutational spectra from bacterial phylogenies and decompose these into specific signatures. We can ascribe some of these signatures to defects in DNA repair pathways, and others to exposure to location-dependent mutagens. We provide examples where these signatures can be used to infer the niche in which bacteria replicate and thereby infer their transmission routes. It is also clear from our data that not all niches are currently associated with specific signatures. Some niches may not contain distinct mutagens that allow discrimination via signature analysis, and in many cases we may not yet have enough data to perform the discrimination analysis. However, we anticipate that further data collection and deeper analyses will allow identification of more signatures at different levels in bacterial phylogenies. This will identify ancestral niches and therefore sources of emergent human pathogens, reveal routes of acquisition of colonization or infection permitting targeted interventions, and provide a mechanism to monitor pathogenic evolution and host adaptation. We envisage that mutational spectra analysis could be applied to viruses and parasites, enabling similar predictions.

## Methods
### Dataset sources and reconstruction of phylogenetic trees
We collated published whole genome sequencing datasets from 84 phylogenetic clades across 31 bacterial species (Supplementary Data 1, accession numbers listed in Supplementary Data 6). Datasets were obtained either from public databases as FASTQ files or genome assemblies, or from study Authors as whole genome sequence alignments or post-recombination removal variable sites alignments (Supplementary Data 1). Where datasets were obtained as genome assemblies, they were initially shredded to FASTQ files containing 100

base pair reads with a 350 base insert size and depth 40 using Fastaq v3.17.0 (https://github.com/sanger-pathogens/Fastaq). Sequencing reads from obtained FASTQ files and shredded assemblies were mapped against a clade-specific reference genome (Supplementary Data 1) using the multiple_mappings_to_bam pipeline v1.6 (https://github.com/sanger-pathogens/bact-gen-scripts) with BWA-MEM as the aligner. Recombination was removed from mapped alignments and whole genome sequence alignments obtained from study Authors using Gubbins v2.4.1[42]. Maximum likelihood phylogenetic trees were reconstructed from post-recombination removal variable sites alignments for all datasets using RAxML v8.2.12[43] with the general time reversible (GTR) model of nucleotide substitution and gamma rate heterogeneity with four gamma classes. Dataset sources are described in more detail in Supplementary Note 1.

### Reconstruction of bacterial mutational spectra
We reconstructed mutational spectra using MutTui v1.1.10 (https://github.com/chrisruis/MutTui) which employs the variable sites alignment, phylogenetic tree and reference genome for each dataset. Mutations are reconstructed onto the phylogenetic tree using treetime v0.8.1[44] which enables identification of the direction of each mutation. It was not possible to identify suitable outgroups to root phylogenetic trees for many datasets and we therefore used midpoint rooted phylogenetic trees. The surrounding nucleotide context (defined as the nucleotide immediately 5′ and nucleotide immediately 3′) of each mutation is inferred from the genome sequence at the start of the respective phylogenetic branch, identified by incorporating substitutions between the root of the tree and the start of the branch into the genome sequence at the root of the tree. We therefore infer the context of each mutation at the time it occurred. The mutational spectrum is constructed by counting the numbers of each contextual mutation across the clade. Single nucleotide mutations are included in the single base substitution (SBS) spectrum, mutations at two adjacent genome positions on the same phylogenetic branch are included in the double

base substitution (DBS) spectrum and tracts of mutations at three or more adjacent genome positions are excluded. To account for differences in G + C content and triplet availability between clades, we rescale each spectrum by dividing the count of each contextual mutation by the frequency of the starting triplet in the clade reference genome.

For several datasets (Supplementary Data 1), the two phylogenetic branches that diverge immediately from the root of the tree represented a large proportion of the mutations in the dataset. We excluded mutations on these root branches in these cases as, due to the necessity to use midpoint rooted trees, their direction may not be inferred accurately and they would account for a large proportion of mutations in the spectrum. Several datasets exhibited evidence of hypermutator branches (Supplementary Data 1) which were excluded from the main clade SBS spectrum but split into separate SBS spectra based on the mutated gene (described in more detail below).

We used the *M. abscessus* SBS spectra we calculated previously where phylogenetic branches were divided into DCCs and non-DCCs[34]. The *M. kansasii* phylogenetic tree contains both the MKMC clade and non-MKMC branches[39]. We split these into separate SBS spectra by labelling branches in the phylogenetic tree as MKMC or non-MKMC which enables MutTui to extract a separate SBS spectrum for each group. The *Burkholderia pseudomallei* genome contains two chromosomes[28]. We calculated the SBS spectrum of each chromosome separately to enable removal of recombination. Due to very high similarity between SBS spectra from chromosomes one and two in each group (cosine similarity >0.99 in each case), we used the chromosome one SBS spectra in further analyses. The *B. cenocepacia* SBS spectrum includes three epidemic clones whose SBS spectra were calculated separately and combined for further analyses.

Overall SBS spectra were compared using UMAP[45] based on the proportion of each of the 96 contextual mutations in each SBS spectrum. To examine the relationship between phylogenetic relatedness and overall spectrum similarity, we calculated the cosine similarity between all pairs of SBS spectra and split the comparisons into within-species, within-genus but different species, within-phylum but different genus and different phylum. The distributions of cosine similarities were compared between groups using two-way ANOVA with Tukey Honestly Significant Difference (HSD) correction.

We compared the degree of context-specificity between mutation types by calculating the variance of the contextual mutation proportions within each of the six mutation types in each of the 84 SBS spectra. The variance distribution between mutation types was compared using two-way ANOVA with Tukey HSD correction. Individual contexts within a mutation type were inferred to be significantly elevated or reduced if their median proportion within the respective mutation type across the 84 SBS spectra was more than 2.5 times the median absolute deviation outside the median of all context proportions in the mutation type.

### Identification of DNA repair gene mutational signatures
We identified potential hypermutator lineages as very long branch lengths (either terminal branches or internal branches where each downstream branch is long) within phylogenetic trees across the 84 datasets; such lineages were identified in *P. aeruginosa*, *B. cenocepacia* and *M. leprae*. We additionally examined a broader *P. aeruginosa* dataset consisting of 18 sequence types and identified hypermutator lineages in this dataset. For each clonal cluster in this dataset, we compared the ratio of transition mutations to transversion mutations on each branch to the background distribution to identify candidate hypermutator branches (Fisher exact test, padj <0.1). We only included branches in the background distribution that had at most 50 substitutions. The gene likely responsible for the hypermutation in each lineage was inferred through identifying DNA repair genes that exhibit a frameshift, insertion/deletion or nonsynonymous mutation on either

the hypermutator branch or an upstream branch where each of the descendent branches are hypermutators. We identified the effects of these mutations using MutTui v1.1.10 applied independently to the branches containing the mutations.

Where a DNA repair gene was mutated on multiple branches, we calculated the SBS spectrum of the mutant as the mean mutational spectrum across branches. We excluded several *P. aeruginosa* branches that had both *mutS* and *mutL* mutations. In several cases, *mutS* and another DNA repair gene were mutated on the same branch; we here calculated the SBS spectrum of the other DNA repair gene by subtracting the mean *mutS* SBS spectrum from the branch SBS spectrum.

We additionally included two previously identified *M. abscessus* hypermutator lineages that arose within individual chronic pulmonary infections[40]. The genes responsible for the hypermutation and the full set of mutations within the hypermutator lineages were previously inferred[40]. We used MutTui v1.1.10 to identify the surrounding nucleotide context of each mutation from a closely related reference genome[40].

To compare the extracted bacterial DNA repair gene signatures with those previously calculated in human cells, we obtained COSMIC SBS signatures from https://cancer.sanger.ac.uk/signatures/sbs/ (date last accessed 24/06/2022) and gene knockout signatures from https://signal.mutationalsignatures.com/ (date last accessed 24/06/2022). We compared mutational patterns through a regression of the proportions of the 16 contextual mutations within the mutation type that is dominant within the respective gene signatures and applied a Benjamini-Hochberg correction on *p*-values from all comparisons.

### Identification of defective DNA repair signatures in *C. jejuni*
To identify mutations that are likely the result of defective DNA repair in *C. jejuni*, we subtracted the SBS spectrum of *E. coli* lineage 34 from the SBS spectrum of each of the five *C. jejuni* clusters. The elevated mutations were decomposed using the signal R package signature.tools.lib v2.1.2[4] into the set of bacterial DNA repair gene signatures we extracted above.

### DNA polymerase III structure modelling
To compare the structures of DNA polymerase III subunits between *P. aeruginosa* and *B. cenocepacia*, we carried out structural modelling. Protein sequences were obtained for each subunit in each species from UniProt[46] (Supplementary Data 7). Homology models were built using SWISS-MODEL online server[47] for all subunits except gamma, for which a local installation of AlphaFold v2.0[48] was used to build models due to sequence coverage below 95% (Supplementary Data 7). We selected the top scoring models for structural analysis in each case. ChimeraX[49] was used for the calculation of electrostatic surfaces, structural alignment and visualisation of predicted models. Template structures for alignment are shown in Supplementary Data 7.

### Linear models to assess the influence of niche on SBS spectra
We compared the fit of linear models incorporating genus alone with models including genus and niche. Niche was coded to best describe current knowledge of replication niches (Supplementary Data 1, Supplementary Note 1). Datasets where replication niche is currently controversial (*M. abscessus* DCCs and *M. kansasii* MKMC) were excluded from this analysis. Linear models were calculated for each mutation type separately using the proportion of the mutation type as the dependant variable. The improvement in model fit when incorporating niche was assessed using analysis of variance (ANOVA); Benjamini-Hochberg correction was applied to *p*-values to correct for multiple testing. Linear model results are shown in Supplementary Data 8.

We additionally tested the influence of the lung/environmental niches and the URT niche on mutational spectra by coding these niches as a binary yes/no variable and comparing the fit of a linear

model including this variable and genus with a model including genus only. Models were again fitted for each mutation type separately. We examined these niches as they contained independent comparisons across multiple genera. Mutation types identified as significant were examined further by comparing mutation type proportions between clades within the niche and clades not within the niche. Significance of differences was assessed within each genus through a bootstrapping approach where we compared the difference between median mutation type proportions in the observed data with that in 1000 randomisations of the niche labels across samples. While C > G was detected as significant in the lung/environment comparison, further inspection did not identify a consistent direction of change across genera so we excluded this mutation type from further analysis.

### De novo signature extraction

Mutational signatures were extracted from each of 14 datasets containing SBS spectra from multiple clades within a species or genus (Supplementary Data 3). We used SigProfilerExtractor v1.1.0[26] which uses nonnegative matrix factorization (NMF) to split a matrix of mutation counts into underlying matrices of mutational signatures and their activities within each input SBS spectrum. The number of signatures is initially set to one and is increased up to a maximum of 25. We identified the optimal number of signatures for each dataset through comparison of the average signature stability (reflecting how well supported the signatures are within the data), mean sample cosine distance (reflecting how well the signatures fit the input SBS spectra) and individual signature stabilities.

We identified cases where the same mutational signature was extracted from multiple datasets by carrying out a hierarchical clustering of all extracted mutational signatures based on cosine distances (calculated as one minus cosine similarity). Signatures were combined if they clustered at cosine distance <0.05, corresponding to cosine similarity >0.95. Where signatures were combined, the final mutational signature was calculated as the mean of the combined signatures. The majority of combined signatures were extracted from taxonomically-nested datasets, with the exception of Bacteria_SBS15 which was extracted from several non-nested species and genus datasets within Bacillota.

The activity of each signature within each SBS spectrum was calculated as the maximum proportion of mutations assigned by Sig-ProfilerExtractor to the signature within any extraction in which the SBS spectrum was included and the signature was extracted.

To compare the clustering of the extracted signatures with bacterial phylogeny, we constructed a dendrogram representing the phylogenetic relationships between bacterial clades from previous literature[24,28,50–61].

### Testing the impact of pathogen niche on mutational spectra

We compared SBS spectra across lung-infecting and environmental clades of *Mycobacteria* and *Burkholderia* through principal component analysis (PCA) of: 1) proportions of the 96 contextual mutation in SBS spectra, 2) proportions of the six mutation types in SBS spectra and 3) proportions of the 16 contextual mutations within each mutation type. To directly compare the SBS spectra of closely related pairs of lung-infecting and environmental clades, we subtracted the SBS spectrum of the environmental clade from that of the lung-infecting clade. The mutations elevated within each clade were decomposed into potential underlying inputs using signal at https://signal.mutationalsignatures.com/ (date last accessed 24/06/2022)[4]. We decomposed mutations elevated in environmental clades into the full set of Environmental Mutagen Signatures, excluding those associated with drug therapy which are unlikely to operate on environmental bacteria. Known and hypothesised lung-infecting clades were decomposed into the full set of lung signatures.

We carried out a targeted NMF decomposition on the full set of SBS spectra from *Mycobacteria* and *Burkholderia* using SigProfilerExtractor v1.1.0[26]. The presence of six signatures was identified as optimum by SigProfilerExtractor and exhibited high average signature stability (0.96 out of maximum 1) and low mean cosine distance between the input and reconstructed SBS spectra (0.016 corresponding to a mean cosine similarity of 0.984). To determine whether any of these signatures are likely niche-associated, we compared the proportion of mutations assigned to SBS spectra from the lung with SBS spectra from the environment and identified one signature that consistently exhibits a higher proportion within lung SBS spectra. We therefore named this signature Bacteria_Lung1.

We compared the DBS spectra calculated by MutTui v1.1.10 between the skin bacteria *C. acnes* and *S. epidermidis*. The DBS spectra of *S. epidermidis* phylogenetic groups A, B and C were combined for this analysis.

### Analyses of the *M. kansasii* MKMC

We calculated transmission networks for *M. kansasii* isolates defining 'probable' transmission as isolates from different patients that differ by fewer than 20 SNPs, and 'possible' transmission as isolates from different patients that differ by fewer than 38 SNPs, based on previous cutoffs established for *M. abscessus*[27].

For mutational burden analysis, the rooted MKMC clade was extracted from the *M. kansasii* phylogenetic tree. Nucleotide mutations were reconstructed onto the tree using Treetime v0.8.1 and the effect of each mutation was inferred using the gene annotation for the *M. kansasii* reference (accession NC022663.1). The impact of each mutation was assessed in the context of the genome at the start of the respective phylogenetic branch. Only nonsynonymous and nonsense mutations were included in mutational burden testing. We assumed a Poisson distribution of mutational burden per gene and calculated the expected number of mutations in each gene based on the gene length and the total number of mutations across the tree. This was compared with the observed number of mutations to identify genes with significantly more mutations than expected. Benjamini-Hochberg correction was applied to account for the number of tests (equal to the number of genes with at least one mutation across the reference genome) and a false discovery rate of 5% used to identify significant genes. We tested whether the *tetR1*/*tetR2* genes mutate significantly more on internal branches than the remaining genes through a Fisher exact test.

### Reporting summary

Further information on research design is available in the Nature Portfolio Reporting Summary linked to this article.

## Data availability

The datasets generated and analysed during this study have been deposited at https://doi.org/10.5281/zenodo.8435731. All source data, including sequence alignments, phylogenetic trees, reference sequences and mutational spectra are available at https://doi.org/10.5281/zenodo.8435731. Accession numbers of all sequences used in this study are provided in Supplementary Data 6. Human mutational signatures were obtained from the COSMIC database (https://cancer.sanger.ac.uk/signatures/). Source data are provided with this paper.

## Code availability

The MutTui pipeline used to reconstruct pathogen mutational spectra is available at https://github.com/chrisruis/MutTui. Additional custom scripts used for data analysis are available at https://doi.org/10.5281/zenodo.8435731.

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

## Acknowledgements

We would like to thank all researchers who helped to obtain published datasets used in this study, including Uzma Basit Khan, Christopher Beaudoin, Sophie Belman, Stephen Bentley, Sebastian Bruchmann, Jessica Calland, Claire Chewapreecha, Jukka Corander, Dorota Jamrozy, Anna Kaarina Pöntinen, Noémie Lefrancq, Stephanie Lo, Neil MacAlasdair, Samuel Sheppard, Andries van Tonder and Lucy Weinert. Funding for this work was provided by The Wellcome Trust through Investigator awards 107032/Z/15/Z (R.A.F., C.R., A.W.) and 200814/Z/16/Z (T.L.B., A.P.P.), Fondation Botnar (Programme grant 6063; R.A.F., J.P., T.L.B., C.R., A.W.) and the UK CF Trust (Innovation Hub Award 001; Strategic Research Centre SRC010; C.R., A.W., T.L.B., R.A.F., J.P.).

## Author contributions

C.R., R.A.F, J.P. conceived and planned the research. C.R., G.T.H designed and implemented mutational spectrum methodology. C.R., A.W., A.P.P., M.M., G.G.R.M., R.C.L analysed, processed and visualised the data. T.B., R.A.F., J.P. jointly supervised the work. C.R., R.A.F., J.P. wrote the first draft of the manuscript. All authors contributed to editing and interpreting the results.

## Competing interests

The authors declare no competing interests.
