## [Peer Review File · Nature Communications]

Mutational spectra are associated with bacterial nicheEditorial Note: This manuscript has been previously reviewed at another journal that is not operating a transparent peer review scheme. This document only contains reviewer comments and rebuttal letters for versions considered at *Nature Communications*.

Reviewer #3 (Remarks to the Author):

This is my third review of the manuscript by Ruis et al. I'd like to thank the authors for their work in revising the manuscript, and for responding to the previous round of comments. The authors have addressed most of my concerns, with two exceptions described in more detail below.

(1) In my previous review, I raised the concern that the statistical significance of the linear model on lines 234-251 could be inflated by phylogenetic correlations below the genus level. To clarify, I was referring to situations where the differences between entire species in the same genus may not be independent because they might correspond to a single niche switch event (or mutational spectrum shift) that occurred in an ancestral branch of the phylogeny. (From the author's response, it sounded like they might have thought that I was referring to the ability to detect niche differences within species, but that was not my primary concern with this comment.)

In the current version of the manuscript, each of the species in the same genus is counted as an "independent" observation in the regression, which can inflate the apparent p-value. The authors mentioned that they considered a phylogenetically aware version of the regression model to correct for this effect, but couldn't figure out a good way to implement it. I understand that decision, but in that case I think it would be necessary to at least highlight this important caveat in the main text, since it has a direct bearing on the p-value that they quote (which is not so low that it will obviously survive such a correction). I don't think that the additional text they added on line 250 is detailed enough for a reader to draw this conclusion.

(2) The authors mentioned that they removed the reference to tobacco smoke per my previous comment, but there is still a reference to smoking on line 274.

We'd like to thank the reviewer for their positive comments.

(1) In my previous review, I raised the concern that the statistical significance of the linear model on lines 234-251 could be inflated by phylogenetic correlations below the genus level. To clarify, I was referring to situations where the differences between entire species in the same genus may not be independent because they might correspond to a single niche switch event (or mutational spectrum shift) that occurred in an ancestral branch of the phylogeny. (From the author's response, it sounded like they might have thought that I was referring to the ability to detect niche differences within species, but that was not my primary concern with this comment.)

In the current version of the manuscript, each of the species in the same genus is counted as an "independent" observation in the regression, which can inflate the apparent p-value. The authors mentioned that they considered a phylogenetically aware version of the regression model to correct for this effect, but couldn't figure out a good way to implement it. I understand that decision, but in that case I think it would be necessary to at least highlight this important caveat in the main text, since it has a direct bearing on the p-value that they quote (which is not so low that it will obviously survive such a correction). I don't think that the additional text they added on line 250 is detailed enough for a reader to draw this conclusion.

We thank the reviewer for this comment and have added an additional caveat on lines 253-254.

(2) The authors mentioned that they removed the reference to tobacco smoke per my previous comment, but there is still a reference to smoking on line 274.

We have removed the reference to tobacco smoke (now line 279).